# Low cost, high performance processing of single particle cryo-electron microscopy data in the cloud

**Michael A Cianfrocco[1,2]\*, Andres E Leschziner[1]**

[1]Department of Molecular and Cellular Biology, Harvard University, Cambridge, United States; [2]Department of Cell Biology, Harvard Medical School, Boston, United States

**Abstract** The advent of a new generation of electron microscopes and direct electron detectors has realized the potential of single particle cryo-electron microscopy (cryo-EM) as a technique to generate high-resolution structures. Calculating these structures requires high performance computing clusters, a resource that may be limiting to many likely cryo-EM users. To address this limitation and facilitate the spread of cryo-EM, we developed a publicly available 'off-the-shelf' computing environment on Amazon's elastic cloud computing infrastructure. This environment provides users with single particle cryo-EM software packages and the ability to create computing clusters with 16–480+ CPUs. We tested our computing environment using a publicly available 80S yeast ribosome dataset and estimate that laboratories could determine high-resolution cryo-EM structures for $50 to $1500 per structure within a timeframe comparable to local clusters. Our analysis shows that Amazon's cloud computing environment may offer a viable computing environment for cryo-EM.

**\*For correspondence:**
mcianfrocco@fas.harvard.edu

**Competing interests:** The authors declare that no competing interests exist.

## Introduction

Cryo-electron microscopy (cryo-EM) has long served as an important tool to provide structural insights into biological samples. Recent advances in cryo-EM data collection and analysis, however, have transformed single particle cryo-EM (*Kuhlbrandt, 2014*; *Bai et al., 2015*), allowing it to achieve resolutions better than 5 Å for samples ranging in molecular weight from the 4 MDa eukaryotic ribosome (*Bai et al., 2013*) to the 170 kDa membrane protein γ-secretase (*Lu et al., 2014*). These high-resolution structures are the result of a new generation of cameras that detect electrons directly without the need of a scintillator, which results in a dramatic increase in the signal-to-noise ratio relative to CCD cameras, the previous most commonly used device (*McMullan et al., 2009*). In addition to direct electron detection, the high frame rate of these cameras allows each image to be recorded as a 'movie', dividing it into multiple frames. These fractionated images can be used to correct for sample movement during the exposure, further increasing the quality of the cryo-EM images (*Campbell et al., 2012*; *Li et al., 2013*; *Scheres, 2014*).

In addition to these technological developments in the detectors, improvements in computer software packages have played an equally important role in moving cryo-EM into the high-resolution era. Atomic or near-atomic structures have been obtained with software packages such as EMAN2 (*Tang et al., 2007*), Sparx (*Hohn et al., 2007*), FREALIGN (*Grigorieff, 2007*), Spider (*Frank et al., 1996*), and Relion (*Scheres, 2012*, *2014*). In general, obtaining these structures involved computational approaches that sorted out the data into homogenous classes that could then be refined to high resolution.

While these advances in microscopy and analysis have been essential for the recent breakthroughs in cryo-EM, their implementation is computationally intensive and requires high-performance

**eLife digest** Microscopes can be used to view objects or structural details that are not visible with the naked eye. A type of microscope called an electron microscope—which uses beams of particles called electrons—is particularly useful for examining tiny objects or structures because it can produce images with a higher level of detail than microscopes that use light.

There are several ways to prepare biological samples for electron microscopy. One technique is called cryo-electron microscopy, or cryo-EM for short, where the sample is rapidly frozen and then viewed under the electron microscope. Using this technique it is possible to produce highly detailed images of viruses, individual compartments within cells and even single proteins.

To convert the images of proteins into three-dimensional models, high-performing clusters of computers are required. It can be difficult and expensive for many scientists to access these resources, which may limit the wider use of cryo-EM in research.

To address this problem and aid the spread of cryo-EM, Cianfrocco and Leschziner developed a publicly available 'off the shelf' system on Amazon's elastic cloud computing infrastructure. This provides users with software packages and the ability to create a cluster containing up to around 480 computers to analyze cryo-EM data.

Cianfrocco and Leschziner tested the system using a publicly available cryo-EM dataset of a structure in yeast cells called the 80S ribosome, which contains proteins and molecules of ribonucleic acid. This revealed that a highly detailed model of the 80S ribosome could be developed in a time frame similar to what it would have taken on a local high-performing computing cluster within a university. The cost of using this system was also competitive in price with that of maintaining a local computing cluster, with the added flexibility of its 'pay-as-you-go' structure.

These findings show that Amazon's cloud computing infrastructure may be a useful alternative to using clusters of computers based within a research institute or university. This will help the spread of cryo-EM as a general tool to reveal the three-dimensional structures of large molecules. Further work is required to make this cloud-based computing tool easily accessible to researchers who may have limited experience with using Linux software and computing clusters.

computing clusters. A recent survey of high-resolution single particle cryo-EM structures showed that refinement of these structures required processing times in excess of 1000 CPU-hours (*Scheres, 2014*). Therefore, computational time (i.e., access to high-performance clusters) may represent a bottleneck to determining high-resolution structures by single particle cryo-EM.

In order to address this limitation, we explored the possibility of using Amazon's elastic cloud computing (EC2) for processing cryo-EM data. To help others take advantage of this resource, we have created a publicly available 'off-the-shelf' software environment that allows new users to start up a cluster of Amazon CPUs preinstalled with cryo-EM software and we have used it to test the performance of Amazon's EC2 platform. We were able to determine a 4.6 Å structure of the 80S ribosome using a published dataset (*Bai et al., 2013*) for an overall cost of $100 USD within a timeframe comparable to that of a local cluster. Given the range of prices for accessing Amazon CPUs (users can bid for significantly reduced costs) and the accessibility statistics, we estimate that typical cryo-EM structures can be determined for $50–$1500 per structure.

## EC2 through Amazon Web Services (AWS)

AWS is a division of Amazon that offers a variety of cloud-based solutions for website hosting and high-performance computing, amongst other services. Many different types of privately held companies take advantage of Amazon's computing infrastructure because of its affordability, flexibility, and security. Of note, global biotechnology companies such as Novartis (*AWS, 2014a*), Bristol-Myers-Squibb (*AWS, 2013*), and Pfizer (*AWS, 2014b*) have utilized the computing power of Amazon for scientific data processing. Many academic researchers have also begun to use Amazon's EC2 resources for analyzing datasets from super-resolution light microscopy (*Hu et al., 2013*), genomics (*Krampis et al., 2012*; *Yazar et al., 2014*), and proteomics (*Mohammed et al., 2012*; *Trudgian and Mirzaei, 2012*).

The overall workflow starts with users logging into a virtual machine ('instance') on AWS (*Figure 1*). AWS offers a variety of instance types that have been configured for different computing tasks. For

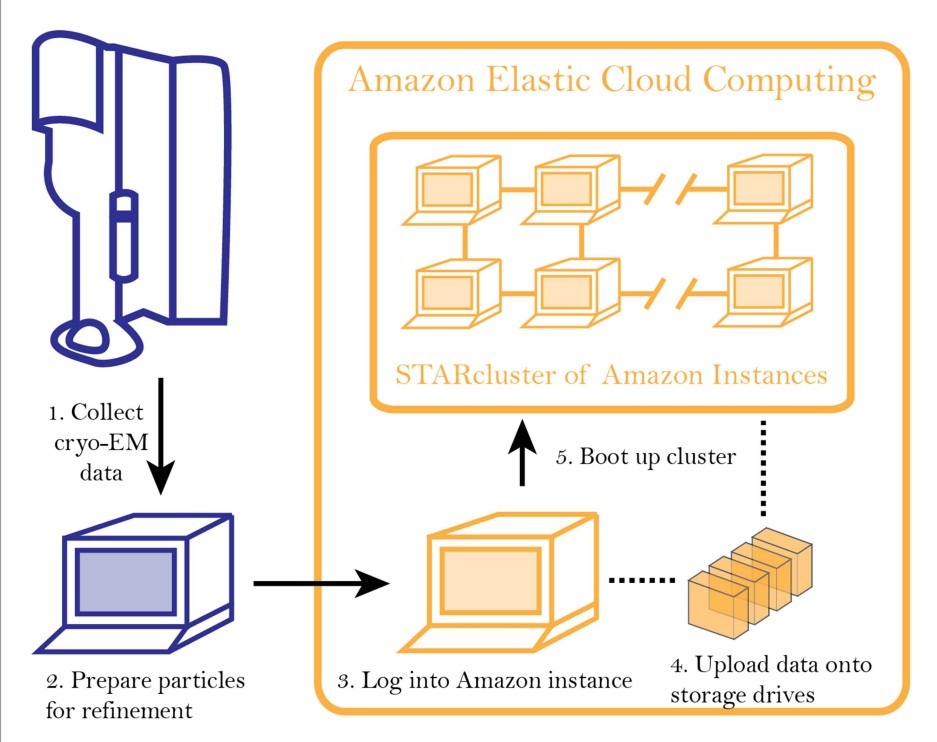

**Figure 1**. Workflow for analyzing cryo-EM data on Amazon's cloud computing infrastructure. After collecting cryo-EM data (Step 1), particles are extracted from the micrographs and prepared for further analysis (Step 2). After logging into an 'instance' (Step 3), data are uploaded to a storage server (elastic block storage) (Step 4). At this point, STARcluster can be configured to launch a cluster of 2–30 instances that is mounted with the data from the storage volume (Step 5). A detailed protocol can be found at an accompanying Google site: http://goo.gl/AlwZJz.

example, instances have been optimized for computing performance, GPU-based calculations, or memory-intensive calculations. After logging onto an instance, storage drives are mounted onto it, allowing data, which can be encrypted for security, to be transferred onto the storage drives (*Figure 1*).

While users can utilize a single instance for calculations, the maximum number of CPU cores per instance is 18. Therefore, creating a computing cluster with a larger number of CPUs on AWS requires additional steps. The Software Tools for Academics and Researchers (STAR) group at the Massachusetts Institute of Technology developed a straightforward package that allows users to group individual AWS instances into a cluster. The STARcluster program is a python-based, open source package that automatically creates a cluster preconfigured with the necessary software to manage a computer cluster (*Ivica et al., 2009*). This package allows users to specify the number of instances to be included in the clusters as well as the instance type. By taking advantage of this tool, private clusters can be built with sizes ranging from 16 to 480 CPUs (*Figure 1*).

## Global availability of spot instances on Amazon EC2

While Amazon provides dedicated access to instances through 'on-demand' reservations, there are 'spot instances' that are 80–90% cheaper than the on-demand price. Spot instances are unused instances within Amazon EC2 that are open for competitive bidding, where users gain access to them by making offers above the current minimum bid. This means that while the on-demand rate for high-memory, 16-CPU instances (called 'r3.8xlarge') is $2.80/hr, spot instance prices can be as low as $0.25–$0.35/hr.

In order to determine if spot instances offer a consistent reduction in price, we analyzed the global availability of r3.8xlarge spot instances. Currently, Amazon has 9 regions worldwide within 7 countries: US-East-1 (United States), US-West-1 (United States), US-West-2 (United States), SA-East-1 (Brazil),

EU-Central-1 (Germany), EU-West-1 (Ireland), AP-Northeast-1 (Japan), AP-Southeast-1 (Singapore), and AP-Southeast-2 (Australia). For each region, we retrieved spot instance prices for r3.8xlarge instances over the past 3 months and analyzed the time they spent at prices below $0.35–$0.65/hr (corresponding to discounts of 87.5–76.8% over the full on-demand rate of $2.80/hr) (*Figure 2* and *Figure 2—figure supplement 1*). This analysis revealed that, globally, 49.8% of r3.8xlarge instances were below $0.35/hr, 12.5% the on-demand price (*Figure 2*). For $0.65/hr, 76.5% below full price, one could access 82.2% of the global r3.8xlarge spot instances. These data indicate that spot instances provide dependable, cost-effective access to Amazon's computing resources.

## Performance analysis of Amazon's EC2 environment with a 80S yeast ribosome dataset

To test the performance of Amazon's EC2 environment, we analyzed a previously published 80S *Saccharomyces cerevisiae* ribosome dataset (*Bai et al., 2013*) (EMPIAR 10002) on a 128 CPU cluster (8 × 16 CPUs; using the r3.8xlarge instance). After extracting 62,022 particles, we performed 2D classification within Relion. Subsequent 3D classification of the particles into four classes revealed that two classes adopted a similar structural state. We merged those two classes and used the associated particles to carry out a 3D refinement in Relion—we were able to obtain a structure with an overall resolution of 4.6 Å (*Figure 3A–C*).

This structure, whose generation included particle picking, CTF estimation, 2D and 3D classification, and refinement, cost us $99.64 on Amazon's EC2 environment. This cost was achieved by bidding on spot instances for particle picking (m1.small at $0.02/hr), 2D classification (STARcluster of r3.8xlarge instances at $0.65/hr), and 3D classification and refinement (STARcluster of r3.8xlarge instances at $0.65/hr). Thus, even though obtaining this structure required 1266 total CPU-hours, Amazon's EC2 computing infrastructure provided the necessary resources to calculate it to near-atomic resolution at a reasonable price.

To further test the performance of Amazon instances, we carried out 3D classification and refinement on a variety of STARcluster configurations using Relion. As before, we ran our tests on clusters of r3.8xlarge high-memory instances (256 GiB RAM and 16 CPUs per instance). Comparing performance across cluster sizes showed that 256 CPUs had the fastest overall time and the highest speedup relative to a single CPU for both 3D classification and refinement (*Figure 4A,B*). However, cluster sizes of 128 and 64 CPUs were the most cost effective for 3D classification and refinement, respectively, as these were the cluster configurations where the speedup per dollar reached a maximum (*Figure 4C*). Importantly, the average time required to boot up these STARclusters was ≤ 10 min for all cluster sizes (*Figure 4D*) and, once booted up, the clusters do not have any associated job wait times. Therefore, these tests showed that Amazon's EC2 infrastructure was amenable to the analysis of single particle cryo-EM data using Relion over a range of STARcluster sizes.

From our analysis of the 80S yeast ribosome, we extrapolated the processing times and combined them with previously published 3D refinement times to estimate typical costs on Amazon's EC2. First, we estimated the cost for

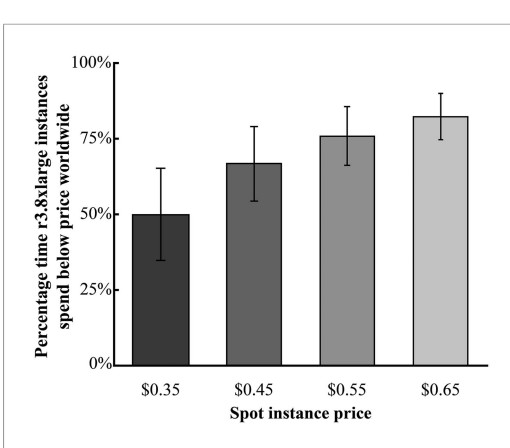

**Figure 2**. Global availability of Amazon r3.8xlarge spot instances. Shown is the average percentage time spent by the r3.8xlarge type of instance when the current spot instance price was less than the queried price. The data are averaged over all Amazon's regions worldwide (except for SA-East-1, which does not offer r3.8xlarge instances). Spot instance prices were calculated over a 90-day period from 1 January 2015—1 April 2015, where the average is shown ± the s.e. Source data: *Figure 2—source data 1*.

The following source data and figure supplement are available for figure 2:

**Source data 1**. Global spot instance price data from 1 January 2015 to 1 April 2015.

**Figure supplement 1**. Availability of virtual machines within regions at specified spot instance prices.

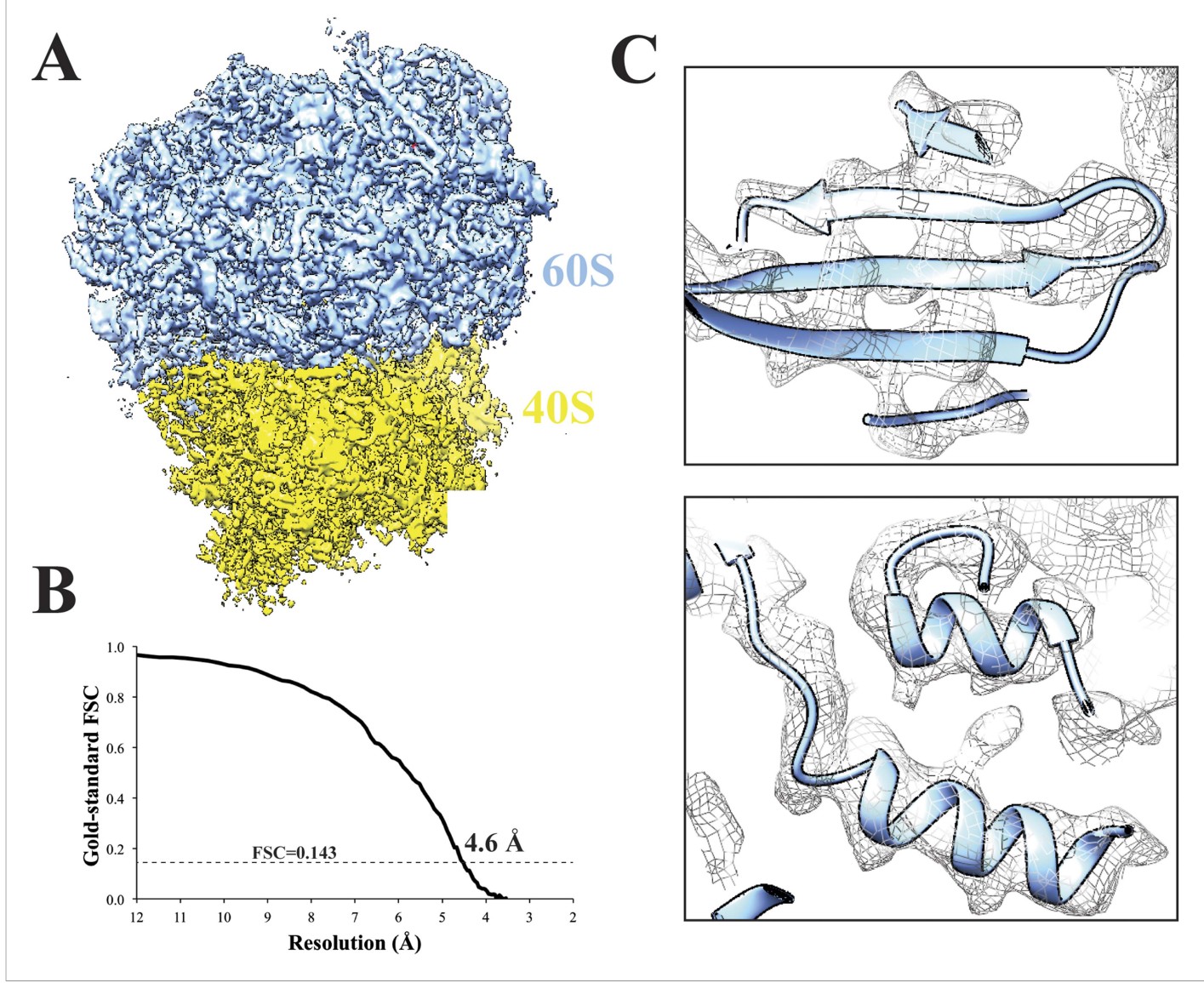

**Figure 3**. Cryo-EM structure of 80S ribosome at an overall resolution of 4.6 Å. (**A**) Overall view of 80S reconstruction filtered to 4.6 Å while applying a negative B-factor of −116 Å². (**B**) Gold standard FSC curve. (**C**) Selected regions from the 60S subunit. Cryo-EM maps were visualized with UCSF Chimera (*Pettersen et al., 2004*). Source data: Dryad Digital Repository dataset (http://datadryad.org/review?doi=doi:10.5061/dryad.9mb54) (*Cianfrocco and Leschziner*).

3D refinement in Relion for previously published structures (*Supplementary file 2A*)—these calculated costs ranged from $12.65 to $379.03 per structure, depending on the spot instance price and required CPU-hours. We then combined these data with conservative estimates for particle picking, CTF estimation, particle extraction, 2D and 3D classification to predict the overall cost of structure determination on Amazon's EC2 (*Supplementary file 2B*). From these considerations, we estimated that published structures could be determined using Amazon's EC2 environment at costs of $50–$1500 per structure (*Supplementary file 2B*).

## EM-packages-in-the-Cloud: a pre-configured software environment for single-particle cryo-EM image analysis

Given the success we had in analyzing cryo-EM data on Amazon's EC2 at an affordable price and within a reasonable timeframe, we have made our software environment publicly available as an

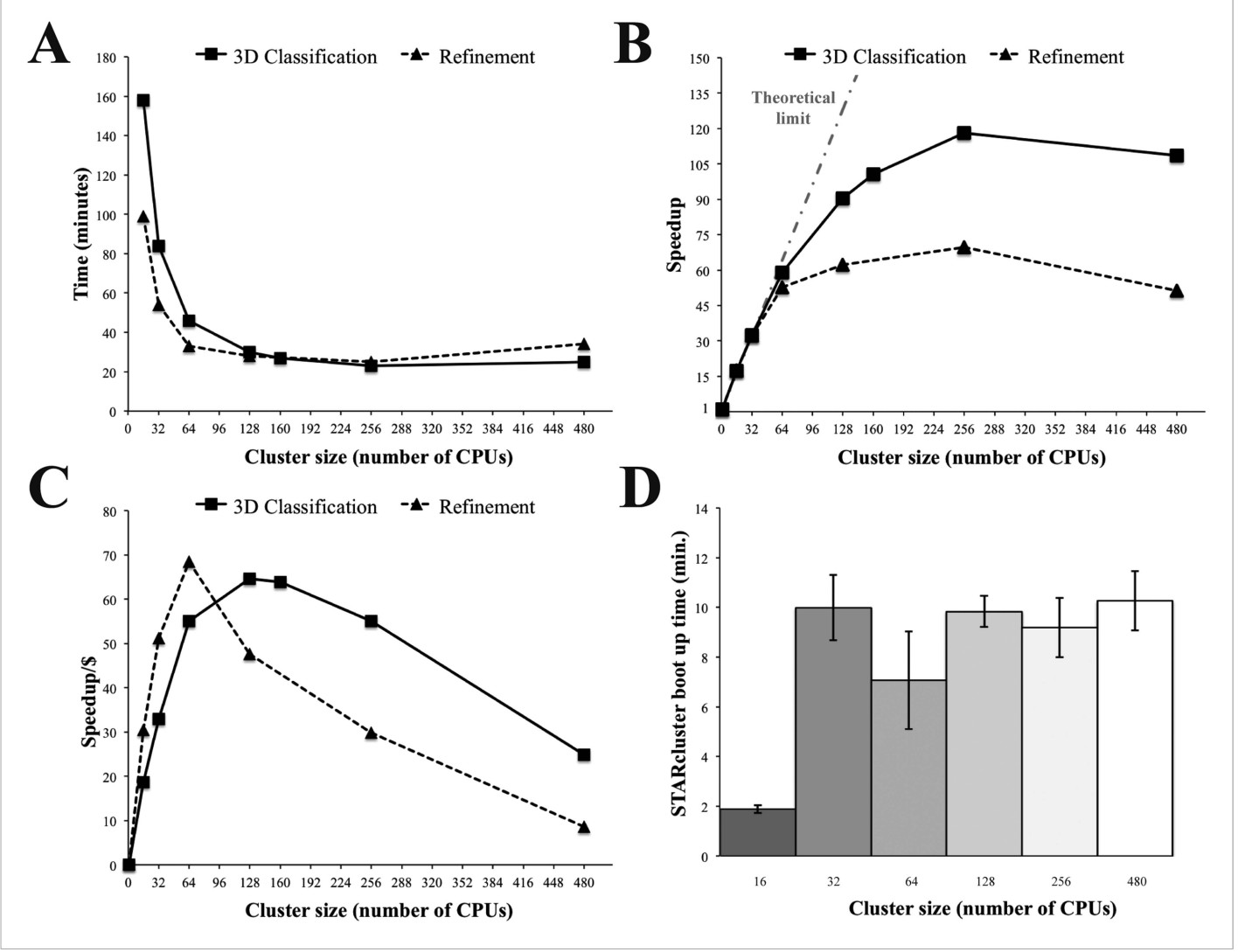

**Figure 4**. Relion performance on STARcluster configurations of Amazon instances. (**A**) Processing times (minutes) for Relion to perform 3D Classification or 3D refinement on 80S ribosome dataset. (**B**) Speedup for each cluster size relative to a single CPU (black line) shown alongside performance estimate for a perfectly parallel cluster using Amdahl's Law (curve labeled 'Theoretical limit'). For cluster sizes ≤ 64 CPUs, Relion exhibits near-perfect performance on STARcluster configurations, while cluster sizes > 64 show that Relion's performance reaches a maximum at 256 CPUs for both 3D classification and 3D refinement. (**C**) Speedup/Cost is plotted against cluster size, where Speedup/Cost is defined as the speedup observed divided by the cost associated with Amazon's pricing at $0.35/hr/16 CPUs. (**D**) Average STARcluster boot up time (± s.d.) was measured for clusters of increasing size (n = 5). Source data: *Figure 4—source data 1*.

The following source data is available for figure 4:

**Source data 1**. Performance analysis statistics for Relion 3D classification and 3D refinement on STARcluster configurations.

'Amazon Machine Image' (AMI), under the name 'EM-packages-in-the-Cloud-v3.93.' The EM-packages-in-the-Cloud-v3.93 AMI provides the software environment necessary for analyzing data on a single instance, and is preconfigured with STARcluster software. The EM-packages-in-the-Cloud-v3.93 AMI has the following cryo-EM software packages installed: Relion (*Scheres, 2012*, *2014*), FREALIGN (*Grigorieff, 2007*), EMAN2 (*Tang et al., 2007*), Sparx (*Hohn et al., 2007*), Spider (*Frank et al., 1996*), EMAN (*Ludtke et al., 1999*), and XMIPP (*Sorzano et al., 2004*). In addition to this AMI that is capable of running on a single instance, we have also made available a second AMI—EM-packages-in-the-Cloud-Node-v3.1—that provides users with the same software packages as

described above, but can set up and run within a cluster of multiple EC2 instances. These two publicly available AMIs allow users to boot up a cluster to analyze cryo-EM data in a few short steps. The protocols describing this can be found as a PDF (*Supplementary file 1*) or on a Google site that is being launched in conjunction with this article: http://goo.gl/AIwZJz. In addition to detailed instructions, the site includes a help forum to facilitate a conversation on cloud computing for single particle cryo-EM.

## Cloud computing as a tool to facilitate high-resolution cryo-EM

Recent advances in single particle cryo-EM have drawn the interest of the broader scientific community. In addition to technical advances in electron optics, the new direct electron detectors and data analysis software have dramatically improved the resolutions that can be achieved for a variety of structural targets. In contrast to the other high-resolution techniques (X-ray crystallography, NMR), structure determination by cryo-EM is extremely computationally intensive. The publicly available 'EM-packages-in-the-Cloud' environment we have presented and characterized here will help remove some of the limitations imposed by these computational requirements.

We believe that cloud-based approaches have the potential to impact the future of cryo-EM image processing in two fronts: (1) new cryo-EM users or laboratories will have immediate access to a high performance cluster, and (2) existing labs may use this resource to increase their productivity. As the number of laboratories using cryo-EM increases, and as existing laboratories begin to pursue high-resolution cryo-EM, gaining immediate access to a high performance cluster may become difficult. For instance, while there are government-funded high performance clusters in the United States (e.g., XSEDE STAMPEDE), it may take up to a month for a user application to be reviewed (Rogelio Hernandez-Lopez, personal communication). Assuming that the application is approved, these clusters may not have appropriate software installed, which further delays data processing. Finally, the user will have a set limit for the number of CPU hours available per project, requiring a new application to be submitted to access the cluster again. All of these problems can be circumvented by using Amazon's EC2 infrastructure, which provides immediate, cost-effective access to hundreds of CPUs with no geographic restrictions.

The power of cloud-based solutions to alleviate the computational burden associated with cryo-EM data processing stems from its high-degree of scalability and reasonable cost. By minimizing computational time and increasing global accessibility, high-performance cloud computing may help usher in the era when high-resolution cryo-EM becomes a routine structural biology tool.

## Materials and methods

### Global availability of spot instances

Global spot instance prices were retrieved from the 90-day period from 1 January 2015 to 1 April 2015 using the Amazon Command Line Tools command *ec2-describe-spot-price-history*. Retrieval of spot instance prices for all regions was implemented automatically in a custom python program *get_spot_histories_all_regions_all_zones.py*. From these spot instance prices, the percentage time spent below given prices was calculated using *measure_time_at_spotPrice.py*, where the cumulative time of spot instances below a given price divided by the total time (90 days). Both programs can be found in the Github repository mcianfrocco/Cianfrocco-and-Leschziner-EMCloudProcessing.

### Setting up a cluster on Amazon EC2 with spot instances

In order to minimize costs, STARclusters were assembled from 'spot instances,' which are unused instances that can be reserved through a bidding process. The spot instances are different from 'on-demand' instances: on-demand instances provide users with guaranteed access while spot instances are reserved until there is a higher bid, at which point the user is logged out of the spot instance. When this happens, the MPI-threaded Relion calculation will abort, requiring the user to resubmit the job to the STARcluster and start Relion from the previous iteration. Even if the user is logged out of all instances within a STARcluster, the data is automatically saved within the EBS-backed volumes on Amazon EC2.

### CPUs vs vCPUs

In selecting an instance type, new users should be aware of the differences between CPUs and vCPUs on Amazon's EC2 network. Namely, that there are two vCPUs per physical CPU on Amazon. This means that while r3.8xlarge instances have 32 vCPUs, there are actually only 16 physical CPU cores in

each instance, with each CPU having two hyperthreads. Practically, this means that Amazon's instances have higher performance than a 16 CPU machine and less performance than a 32 CPU machine. To account for this difference, all numbers reported here were CPU numbers that were converted from vCPUs: 1 CPU = 2 vCPUs.

## Image processing

Micrographs from the 80S *S. cerevisiae* ribosome dataset (*Bai et al., 2013*) were downloaded from the EMPIAR database for electron microscopy data (EMPIAR 10002). The SWARM feature of EMAN2 (*Tang et al., 2007*) was used to pick particles semi-automatically. Micrograph defocus was estimated using CTFFIND3 (*Mindell and Grigorieff, 2003*). The resulting particle coordinates and defocus information were used for particle extraction by Relion-v1.3 (*Scheres, 2012*, *2014*). The particle stacks and associated data files were then uploaded to an elastic block storage volume on Amazon's EC2 processing environment at a speed of 10 MB/s (24 min total upload time).

After 2D classification in Relion, 3D classification was performed on 62,022 80S Ribosome particles (1.77 Å/pixel), also in Relion. These were classified into 4 groups (T = 4) for 13 iterations using a ribosome map downloaded from the Electron Microscopy Data Bank (EMDB-1780) that was low pass filtered to 60 Å. Further 3D classification using a local search of 10° and an angular sampling of 1.8° continued for 13 iterations. At this point, two classes were identified as belonging to the same structural state and were selected for high-resolution refinement (32,533 particles). Refinement of these selected particles continued for 31 iterations using *3D auto-refine* in Relion. The final resolution was determined to be 4.6 Å using *Post process* in Relion, applying a mask to the merged half volumes and a negative B-factor of −116 Å$^2$.

## Performance analysis

80S ribosome data were reanalyzed on clusters of increasing size using both 3D classification and 3D refinement. The time points collected involved running 3D classification for 2 rounds and 3D refinement for 6 rounds, using the same number of particles and box sizes listed above: 62,022 particles for classification and 32,533 particles for refinement with box sizes of 240 × 240 pixels. The Relion commands were identical to the commands used above and the calculations were terminated after the specified iteration.

From these time points, the speedup of each cluster size was calculated relative to a single CPU. Speedup (*S*) was calculated as:

$$S = \frac{\text{Calculation time for 1 CPU}}{\text{Calculation time for } x \text{ CPUs}}.$$

The measured speedup values were then compared to the speedup expected for a perfectly parallel algorithm (*P* = 1) using Amdahl's law (*Amdahl, 1967*):

$$S = \frac{1}{(1-P) + \frac{1}{n}(P)} = \frac{1}{(1-1) + \frac{1}{n}(1)} = n,$$

Where *P* is the fraction of an algorithm that is parallel and *n* is the number of processors. The calculation times for 3D classification on a single CPU were obtained by using 1 CPU on a 16 CPU r3.8xlarge instance. For calculating a 3D refinement on a single CPU, (or two vCPUs), the refinement was run on 4 vCPUs and then converted to a single CPU (or two vCPUs) by multiplying the calculation time by 2. For cost analysis, the measured speedup was divided by the cost to run the job on spot instances of r3.8xlarge at a price of $0.35/hr. Cluster boot up times were calculated from the elapsed time between submitting the STARcluster command and the STARcluster fully booting up.

## Data accession information

Further information regarding 'EM-Packages-in-the-Cloud' can be found in *Supplementary file 1* and at an associated Google Site: http://goo.gl/AIwZJz. The final 80S yeast ribosome structure at 4.6 Å has been submitted to the EM Databank as EMDB 2858. A detailed description of global spot instance price analyses and image processing is available at https://github.com/mcianfrocco/Cianfrocco-and-Leschziner-EMCloudProcessing/wiki. Associated computing scripts and data files have been uploaded to Github (https://github.com/mcianfrocco/Cianfrocco-and-Leschziner-EMCloudProcessing) and Dryad

Digital Repository (http://dx.doi.org/10.5061/dryad.9mb54) (*Cianfrocco and Leschziner, 2015*), respectively.

## Acknowledgements

We would like to thank all the members of the Leschziner and Reck-Peterson labs for critical discussions. We would like to especially thank Rogelio Hernandez-Lopez, Anthony Roberts, and Daniel Cianfrocco for critical feedback on the development of this Amazon computing environment. We also would like to thank the Structural Biology Consortium (SBGrid) for pricing information on cluster and file server sizes. MAC is an HHMI fellow of the Damon Runyon Cancer Research Foundation and AEL is supported by NIH/NIGMS (R01 GM107214 and R01 GM092895A).

## Additional information

### Funding

| Funder | Grant reference | Author |
|---|---|---|
| Damon Runyon Cancer Research Foundation (Damon Runyon) | 2171-13 | Michael A Cianfrocco |
| National Institutes of Health (NIH) | R01GM107214 | Andres E Leschziner |
| National Institutes of Health (NIH) | R01GM092895A | Andres E Leschziner |

The funders had no role in study design, data collection and interpretation, or the decision to submit the work for publication.

### Author contributions

MAC, Conception and design, Acquisition of data, Analysis and interpretation of data, Drafting or revising the article; AEL, Analysis and interpretation of data, Drafting or revising the article

## Additional files

### Supplementary files

• Supplementary file 1. Step-by-step tutorial describing how to use Amazon's EC2 environment to analyze cryo-EM data.

• Supplementary file 2. Comparison of estimated processing times and costs for recent near-atomic cryo-EM structures on Amazon's EC2. Source data: *Supplementary File 3*.

• Supplementary file 3. Source data for tables in *Supplementary file 2*.

### Major datasets

The following dataset was generated:

| Author(s) | Year | Dataset title | Dataset ID and/or URL | Database, license, and accessibility information |
|---|---|---|---|---|
| Cianfrocco MA, Leschziner AE | 2014 | Data from: Single particle cryo-electron microscopy image processing in the cloud: High performance at low cost | http://dx.doi.org/10.5061/dryad.9mb54 | Available at Dryad Digital Repository under a CC0 Public Domain Dedication. |

The following previously published dataset was used:

| Author(s) | Year | Dataset title | Dataset ID and/or URL | Database, license, and accessibility information |
|---|---|---|---|---|
| Bai XC, Fernandez IS, McMullan G, Scheres SH | 2013 | S.cereviseae 80S ribosome direct electron detetector dataset | http://www.ebi.ac.uk/pdbe/emdb/empiar/entry/10002/ | Publicly available at Electron Microscopy Pilot Image Archive. |

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
