## [Decision Letter]

Thank you for sending your work entitled “Low cost, high performance processing of single particle cryo-electron microscopy data in the cloud” for consideration at *eLife*. Your Tools and Resources article has been favorably evaluated by John Kuriyan (Senior editor) and three reviewers, one of whom is a member of our Board of Reviewing Editors.

The following individuals responsible for the peer review of your submission have agreed to reveal their identity: Sjors Scheres (Reviewing editor); Steven Ludtke (peer reviewer). A further reviewer remains anonymous.

The Reviewing editor and the other reviewers discussed their comments before we reached this decision, and the Reviewing editor has assembled the following comments to help you prepare a revised submission.

All three reviewers agreed that this paper represents a novel and original way of alleviating the high computational burdens that many cryo-EM labs face with the advent of huge amounts of data from new direct electron detectors. As this paper has the potential to accelerate discovery, and to change how many labs operate, publication was recommended by all three.

The following concerns (in order of importance) were raised:

Firstly, one of the reviewers had actually recently performed the EC2 cost analysis himself and was surprised to see the estimates in this paper less expensive than his own. He writes: “The first issue is an Amazon trick. The r3.8xlarge instances are marketed as “32 vCPUs”. Actually this is 32 threads running on 16 cores when you read the fine print. For image processing this is generally 18-20 “CPUs” worth of compute power for the “32 vCPUs”. CPU-hr/mo/yr normally measure core-hours, not thread hours. This is a factor of ∼1.8. If the authors disagree, I would encourage them to run a scalability test on a small problem on a single 32 vCPU instance. Perusing the Amazon site, the cost of a single r3.8xlarge instance with 16 physical cores is currently $2.80 for on demand use, rather than the $0.35 quoted by this manuscript. While it is possible to reduce this by up to ∼50% through contract prepurchase, the only mechanism I can see for getting the price anywhere close to the cited level is by bidding on unused hours, which can mean substantial delays. Currently the purchase price for an equivalent cluster is ∼$350/core ($175/thread), or ∼$7500 for a node almost identical to the $2.80/hr instance. Anyway, by my calculation, when you take all expenses into account, EC2 is about 3-5x more expensive than owning a cluster. However, if the Amazon price were suddenly 10x lower, this would be compelling. If my cost analysis is in error, I would be honestly grateful to see a correction, as it would substantially alter how we operate.”

Secondly, in Table 1, the reported times for the other 3 cases are incorrect. They are merely the same of the values reported for the old and the new movie processing in the [22]
*eLife* paper. There are no reported CPU costs for the entire processing procedures of these structures in the literature. However, on the RELION wiki (http://www2.mrc-lmb.cam.ac.uk/relion/index.php/FAQs#Computational_issues) it is stated: “We do 3.x Angstrom ribosome reconstructions from say 100-200 thousand particles in approximately two weeks using around 200-300 cores in parallel”. This would result (given current cost estimates) in about $800 per structure, which is still very reasonable.

Thirdly, the authors are encouraged to base their analysis on Amdahl's Law instead of the “near-linear increase” stated in the paper. While I would normally consider this a minor concern, this analysis will yield numbers which will be interesting to consider.

[Editors’ note: the decision letter after resubmission follows.]

Thank you for choosing to send your work entitled “Low cost, high performance processing of single particle cryo-electron microscopy data in the cloud” for consideration at *eLife*. Your revised submission has been evaluated by John Kuriyan (Senior editor) and Sjors Scheres (Reviewing editor). Based on our discussions and the individual reviews sent previously, we regret to inform you that your work will not be considered further for publication in *eLife*.

It is unfortunate that the standard prices for Amazon's cloud are so high. We feel that bidding on unused hours is likely to be unpredictable in the future and this lack of predictability makes it less appropriate as a means of evaluating the costs of the calculations described in the paper. The true cost of doing an entire structure determination project (not only a single refinement run) at standard Amazon prices would probably quite substantially higher than the value mentioned in the Abstract. We fear that this could more expensive than buying a local cluster. Although some smaller labs may still benefit from the cloud setup, this paper will not likely change the way cryo-EM labs work in general.

[Editors’ note: after an appeal against the decision, further revisions were requested before acceptance.]

In order to give a fair and transparent view to the casual reader, we feel that the addition of a discussion on the costs of a “typical” structure determination project would add value to the paper. The phrase “as we illustrate here by determining a near-atomic resolution structure of the 80S yeast ribosome for $28.89 USD in ∼10 hours” in the current Abstract is not representative of a typical case. Many data sets will contain several hundreds of thousands of particles, and each 2D or 3D classification or refinement run will cost in the order of 100-200$ each (based on your estimates for gamma-secretase and mitoribosome). As in a typical project one would run multiple of these jobs, real costs will quickly reach more than a thousand dollars per structure, even when using the $0.35/hour bidding rate. This is perfectly well acceptable and still competitive with buying a local cluster. But discussing such values in the paper will prevent unpleasant surprises when PIs start receiving EC2 bills.

---

## [Author Response]

*The following concerns (in order of importance) were raised*:

*Firstly, one of the reviewers had actually recently performed the EC2 cost analysis himself and was surprised to see the estimates in this paper less expensive than his own. He writes:* “*The first issue is an Amazon trick. The r3.8xlarge instances are marketed as* “*32 vCPUs*”*. Actually this is 32 threads running on 16 cores when you read the fine print. For image processing this is generally 18-20* “*CPUs*” *worth of compute power for the* “*32 vCPUs*”*. CPU-hr/mo/yr normally measure core-hours, not thread hours. This is a factor of ∼1.8. If the authors disagree, I would encourage them to run a scalability test on a small problem on a single 32 vCPU instance*.

We would like to thank the reviewers for pointing out the difference between vCPUs and CPUs. To make sure there is no confusion, we updated the text so that all analysis and discussion of Amazon EC2 involves a discussion of CPUs and CPU core-hours (instead of hyperthreads and hyperthread-hours). This will allow the manuscript to be readily compared to other published work that reports on CPUs and CPU core-hours. We also included a section within the Materials and methods where we discuss the differences between CPUs and vCPUs.

*Perusing the Amazon site, the cost of a single r3.8xlarge instance with 16 physical cores is currently $2.80 for on demand use, rather than the $0.35 quoted by this manuscript. While it is possible to reduce this by up to ∼50% through contract prepurchase, the only mechanism I can see for getting the price anywhere close to the cited level is by bidding on unused hours, which can mean substantial delays. Currently the purchase price for an equivalent cluster is ∼$350/core ($175/thread), or ∼$7500 for a node almost identical to the $2.80/hr instance. Anyway, by my calculation, when you take all expenses into account, EC2 is about 3-5x more expensive than owning a cluster. However, if the Amazon price were suddenly 10x lower, this would be compelling. If my cost analysis is in error, I would be honestly grateful to see a correction, as it would substantially alter how we operate*.”

We agree that Amazon EC2 is not particularly cost effective for the scenarios presented by the reviewers. We were able to minimize costs by bidding on unused hours through spot instance requests. This allowed us to reserve instances at a price of $0.35/hr instead of the on-demand cost of the r3.8xlarge is $2.80/hr. Also, we have not seen any delays in getting access to these spot instances (Figure 2), which has made them reliable during our experience with EC2. To help convey these ideas to the reader, we have included a section within the Materials and methods section describing how we were able to reserve the spot instances, comparing them to the on-demand instances.

*Secondly, in Table 1, the reported times for the other 3 cases are incorrect. They are merely the same of the values reported for the old and the new movie processing in the*
[22] eLife *paper. There are no reported CPU costs for the entire processing procedures of these structures in the literature. However, on the RELION wiki (**http://www2.mrc-lmb.cam.ac.uk/relion/index.php/FAQs#Computational_issues**) it is stated:* “*We do 3.x Angstrom ribosome reconstructions from say 100-200 thousand particles in approximately two weeks using around 200-300 cores in parallel*”*. This would result (given current cost estimates) in about $800 per structure, which is still very reasonable*.

During the preparation of the manuscript, we overlooked this incorrect comparison and we would like to thank the reviewers for raising this issue. We have updated the table to indicate that we are comparing 3D refinement processing times, not total processing times.

*Thirdly, the authors are encouraged to base their analysis on Amdahl's Law instead of the* “*near-linear increase*” *stated in the paper. While I would normally consider this a minor concern, this analysis will yield numbers which will be interesting to consider*.

We performed the suggested analysis and found it to be very helpful in providing a theoretical limit to increases expected by Amazon STARcluster configurations. The results have been included in Figure 2.

[Editors’ note: the author responses to the decision on the revised submission follows.]

While we understand the rationale behind your decision, we believe that some assumptions underlying this decision are not entirely correct. We realize we are at fault for not having presented enough data in our manuscript on the likelihood of being able to secure computational resources at the costs we quoted. We have included below these data, as part of a detailed response to the main points raised in your decision letter. We hope that this new information will more successfully make the point that cloud-based computing represents a cost-effective and general solution to the computational burden imposed by cryo-EM data analysis.

*It is unfortunate that the standard prices for Amazon's cloud are so high. We feel that bidding on unused hours is likely to be unpredictable in the future and this lack of predictability makes it less appropriate as a means of evaluating the costs of the calculations described in the paper*.

We would like to thank you for bringing this up because it is an important concern that we should have addressed more clearly. As pointed out, the prices of Amazon’s instances can change. Despite this, the vast majority of spot instance prices have remained consistently low. This can be seen in an analysis of r3.8xlarge spot instance prices over the past three months in both the United States (Virginia) and European Union (Ireland) (Figure 5). This analysis showed that users can gain access to > 90% of instances at a price of $0.65/hr, which is 25% of the standard rate of $2.80/hr. Even at a price of $0.65/hr, the cost of 3D classification and refinement of the 80S ribosome would have been $53.64 (Table 1).

Author response image 1.Percentage of instances below bid price over last 90 days. Shown are the percentages of r3.8xlarge instances that are below the spot instance price across different regions and zones.**DOI:**
http://dx.doi.org/10.7554/eLife.06664.015

Author response table 1.Cost of 80S ribosome 3D classification and refinement on a 128 CPU STARcluster configuration at increasing spot instance price.**DOI:**
http://dx.doi.org/10.7554/eLife.06664.016Spot instance bid price$0.35$0.45$0.55$0.6580S ribosome 3D classification and refinement cost$28.89$37.14$45.39$53.64

We would also like to make two final points regarding the future of prices for Amazon instances:

Competition: Multiple cloud-based service providers (e.g. Google, Microsoft, Alibaba) already provide virtual machines for users across the world. These providers are vying to supply large privately held companies with computing resources. This will translate into stabilizing market forces that will help keep the price of cloud computing stable (if not lower it).

Moore’s law: Further development of CPU processing power will make the most ‘powerful’ generation of CPUs on Amazon cheaper within 2 years, further driving down cost.

*The true cost of doing an entire structure determination project (not only a single refinement run) at standard Amazon prices would probably quite substantially higher than the value mentioned in the Abstract. We fear that this could more expensive than buying a local cluster*.

We would also like to thank the reviewers for raising the issue of the ‘true cost’ of single particle analysis on Amazon EC2. The estimates outlined below will highlight the cost-effectiveness of Amazon’s EC2 infrastructure; individual virtual machines can be used at rates as low as $0.003/hr. The following is an estimate for the entire processing of a single dataset:

1) Picking particles and CTF estimation: Estimated cost $2.00 (100 hrs at $0.02/hr)

For this estimate, we used a spot instance price of $0.02/hr on an m1.small instance, which has had an average spot instance price of $0.0103 +/- 0.0038 over the last 90 days in US-Virginia. At this rate, a user could select particles and estimate the CTF over the course of 100 hours (2.5 x 40 hr. work weeks) for $2.00.

2) Particle extraction: $0.20 (10 hrs at 0.02/hr)

The particle extraction of the 80S ribosome presented in our manuscript required ∼10 hrs.

3) 2D classification using Relion: $23.29 (8.32 hrs at $2.80/hr)

For the 80S ribosome dataset presented in [3], we performed 2D classification using Relion on a STARcluster of 128 CPUs at price of $0.35/instance/hr on 8 instances at a total cost of $2.80/hr. The final cost for classification was $23.29 to classify 62,022 at a pixel size 3.54 and a box size of 120 x 120 pixels into 250 classes (T=2) over 25 iterations.

4) 3D classification & refinement using Relion: $28.89 (10.32 hrs at $2.80/hr)

This cost reflects of price of $0.35/hr/instance on a 128 CPU STARcluster, which comprises eight r3.8xlarge instances.

In summary:

Particle picking & CTF estimation $2.00

Particle extraction $0.20

2D classification using Relion $23.29

3D classification & refinement using Relion $28.89

Total cost $54.38

*Although some smaller labs may still benefit from the cloud setup, this paper will not likely change the way cryo-EM labs work in general*.

The spread of cryo-EM as a common structural biology tool necessitates the spread of a cryo-EM computing software platform. Unlike X-ray crystallography, which can be used to solve atomic structures on a single workstation, cryo-EM data analysis inherently requires access to hundreds of CPUs for 3D structure determination. Currently, major universities have invested heavily in computing infrastructures, which are absent from many universities around the world. Therefore, as more labs begin to use cryo-EM software, they will need access to larger computing resources. Inherently, many of those newcomers may be small, or there may only be limited numbers of members within a laboratory that need high performance computing. We believe cloud-based computing will provide a solution to these challenges.

While cloud computing is continuing to mature, we have created a tool that will immediately address the computational burden imposed by cryo-EM data analysis. Cloud-based computing is a tool that could and would scale rapidly with the spread of cryo-EM.